# A Survey on Verifiable Cross-Silo Federated Learning

**Aleksei Korneev**                                                        *aleksei.korneev@inria.fr*
*University of Lille, Inria,*
*CNRS, Centrale Lille,*
*UMR 9189 - CRIStAL*

**Jan Ramon**                                                              *jan.ramon@inria.fr*
*University of Lille, Inria,*
*CNRS, Centrale Lille,*
*UMR 9189 - CRIStAL*

**Reviewed on OpenReview:** *https: // openreview. net/ forum? id= uMir8UIHST*

## Abstract

Federated Learning (FL) is a widespread approach that allows training machine learning (ML) models with data distributed across multiple storage units. In cross-silo FL, which often appears in domains like healthcare or finance, the number of participants is moderate, and each party typically represents a well-known organization. For instance, in medicine data owners are often hospitals or data hubs which are well-established entities. However, malicious parties may still attempt to disturb the training procedure in order to obtain certain benefits, for example, a biased result or a reduction in computational load. While one can easily detect a malicious agent when data used for training is public, the problem becomes much more acute when it is necessary to maintain the privacy of the training dataset. To address this issue, there is recently growing interest in developing verifiable protocols, where one can check that parties do not deviate from the training procedure and perform computations correctly. In this paper, we present a survey on verifiable cross-silo FL. We analyze various protocols, fit them in a taxonomy, and compare their efficiency and threat models. We also analyze Zero-Knowledge Proof (ZKP) schemes and discuss how their overall cost in a FL context can be minimized. Lastly, we identify research gaps and discuss potential directions for future scientific work.

## 1 Introduction

Nowadays, the broad propagation of machine learning (ML) technologies is rapidly increasing. ML applications affect a variety of fields such as medicine, finance, marketing, education, and many others (Garg & Mago, 2021; Ozbayoglu et al., 2020; Herhausen et al., 2024; Pinto et al., 2023). Many ML approaches rely on a process of training on historical data: a model learns statistical patterns that later allow new predictions to be inferred. However, in some cases, data may contain private or confidential information; therefore, access to such data is limited, and applying ML must be done with extreme caution, either due to an interest in privacy of the data owners (DOs), e.g., individual persons caring about their privacy or companies caring about intellectual property, or for regulatory compliance, e.g., with the General Data Protection Regulations (GDPR).

In federated learning (FL), multiple DOs, who are also referred to as clients, can train a model together, possibly under the coordination of a central server, by exchanging encrypted messages without revealing their private data. As a result, researchers can benefit from a large amount of shared data and at the same time preserve privacy. However, while preserving privacy in FL allows protecting sensitive information, it also generates an additional challenge in the verification of participants' behavior. Indeed, due to the possibility of malicious actions, it is important to ensure that all calculations are performed correctly even if the used data is private.

FL is often divided into two categories: cross-device and cross-silo. In the cross-device FL setting, data comes from a large number of small and usually anonymous devices with low computational capacities. Anonymity complicates penalizing clients; a single client is free to abort or violate the procedure at any time. In contrast, in this paper we focus on cross-silo FL where the number of parties is moderate; each party is usually a well-known and large entity that is expected to cooperate in the entire training process via devices with high computing power. Each party has an incentive to care about its reputation and can be held liable if it is found to be fraudulent. For instance, cross-silo FL appears in the healthcare domain, where DOs are medical centers or hospitals that collaborate to train ML models to improve patient care. Although the need for countermeasures against malicious attacks in such setting could be reduced due to the liability of participants, verification of the calculations is still required to establish confidence in FL's performance.

Recently, dozens of research works devoted to verifiable FL have been published, proposing methods to ensure the verifiability of the parties' computations, using different infrastructures and relying on various assumptions. Nonetheless, to the best of our knowledge, verifiability in the context of cross-silo FL has not been thoroughly studied. Huang et al. (2022a) studied challenges of cross-silo FL setting in details, but the verifiability property was not taken into account. In (Zhang & Yu, 2022; Tariq et al., 2023) authors were focused on verifiability in FL, however features of the cross-silo setting were not considered and protocols' efficiency was not analyzed. In contrast to the taxonomy presented by (Zhang & Yu, 2022), our taxonomy is based on the verification techniques, which allows grouping similar approaches together and identify common design patterns that directly impact protocol properties (e.g., complexities, threat model assumptions). Unlike Tariq et al. (2023), we formally describe the underlying verification techniques and analyze the complexities and threat models of the approaches, providing a deeper level analysis. Mohamad et al. (2023) presented a SoK paper devoted to secure aggregation protocols and included verification in the list of challenges, nevertheless, specific features of cross-silo FL were not in the scope of the paper and an efficiency analysis of verification techniques was not performed. Lastly, in (Labs, 2023; Xing et al., 2023) authors studied applications of various Zero-Knowledge Proof (ZKP) schemes for ML, but these works do not address FL.

In this paper, we provide a survey on verifiable cross-silo FL. Our contributions are summarized as follows:

- to the best of our knowledge, we are the first to conduct a survey on verifiable FL while studying specific challenges of the cross-silo setting;

- we propose a new taxonomy of existing verifiable cross-silo FL protocols while analyzing their efficiency and threat models;

- we perform a comparison of ZKP schemes from the perspective of applying them in cross-silo FL and discuss the influence of parameters such as the ZKP scheme and circuit size on the cost;

- we define future challenges and identify research gaps.

The rest of the paper is organized in the following way: Section 2 introduces some background and notations of FL and crpytographic primitives; Section 3 presents an analysis of existing verifiable cross-silo FL protocols; Section 4 is devoted to a comparison of ZKP schemes from the perspective of their applicability for cross-silo FL, then we describe a storage cost optimization for ZKP-based FL protocols; Section 6 describes challenges and research gaps; and Section 7 concludes the paper.

## 2 Background

### 2.1 FL process

The majority of FL approaches contain two main categories of operations: local operations (both on the side of the clients and the server) and aggregation of clients' values. Additionally, there are also other operations specific to certain FL protocols, for example, where participants should draw random numbers, exchange cryptographic keys or select a subset of parties to communicate with.

In this paper, we consider both settings where the aggregation is coordinated or performed by a central server and settings where DOs perform the aggregation in a decentralized way.

Some ML algorithms involve running an optimization algorithm, e.g., stochastic gradient descent (SGD). We refer to each iteration of such algorithm as an epoch.

## 2.2 Cross-silo FL properties

While analyzing the suitability of various algorithms for the cross-silo FL setting, we assume that:

1. the number of participants is moderate (at most several thousands);

2. all participants have an incentive to care about their reputation, they may only cheat in a way which can not be detected by others;

3. all participants agree on the model to be trained (type of calculations to be executed);

4. DOs possess computationally sufficiently powerful equipment.

In this survey, we do not focus on the cross-device setting due to its unique characteristics (e.g., a large number of parties, typically low computational resources) that would require a separate analysis to properly compare protocols with each other and determine the suitability of verification techniques.

## 2.3 Adversarial attacks on FL

In a survey (Rodríguez-Barroso et al., 2023) on FL threats two classes of adversarial attacks were distinguished:

- privacy attacks, whose purpose is to infer sensitive information from the learning process;

- attacks which aim at modifying the behavior or output of the FL process.

The development of a FL protocol that is resistant to adversarial attacks requires applying a combination of various privacy enhancing technologies (PETs). For example, in order to prevent privacy attacks, authors of state-of-the-art solutions employ differential privacy (DP), multi-party computation (MPC), secure shuffling and Trusted-Execution Environment (TEE) among others.

In this paper, we study verification techniques that allow mitigating the second class of attacks, attacks on the federated learning process, such as when an adversary intentionally uses incorrect data (data poisoning) or performs computations incorrectly (model poisoning) to bias the resulting model. Authors of the considered papers applied commitment schemes, homomorphic hash functions, ZKP schemes and other methods to ensure that the federated model is computed correctly. A detailed analysis of these methods is presented in sections 3 and 4.

## 2.4 Verifiable FL

In the scope of this paper, we rely on a definition of Verifiable FL inspired by Zhang & Yu (2022):

*Definition (Verifiable FL).* FL is verifiable if selected parties are able to verify that the tasks of all participants are correctly performed without deviation.

In practice, it is often assumed that a subset of participants follows the protocol honestly, while others are treated as potentially malicious. Depending on the specific use case, various actors within the FL, such as the server, aggregator, or DOs, could be considered untrusted, either individually or together. If only a fraction of tasks is verified, others are either assumed to be performed honestly or remain vulnerable to attacks.

We note that the definition does not specify the exact phase of the FL process at which the verification process occurs, nor how the verifiers are selected. These details depend on the concrete design choices, assumptions, and other features of a particular verifiable protocol. In practice, it is common for FL participants to also act as verifiers. In such cases, the verification algorithm is executed after receiving potentially malformed data from interactions with untrusted parties. For example, clients may verify that the received updated global model has been correctly computed by the aggregator. However, there exists alternative scenarios: for instance, some approaches allow any external party to act as a verifier by inspecting all publicly available proofs. For such protocols, verification may occur at any moment, even when the FL training is finished.

Following this definition, in contrast to the survey by Tariq et al. (2023), we only include approaches that at least partly verify computations of the FL process. For example, we do not analyze protocols which are focused only on verification of identity, ownership, or data provenance. We also exclude protocols considered in (Mohamad et al., 2023) that aim to prevent model poisoning attacks only by analyzing distribution of values submitted by parties. Such methods efficiently mitigate some attacks, but do not allow one to verify the correctness of individual computations or of the individual uses of the input data, e.g., an individual outlier input value is infrequent but possibly valid. Moreover, their efficiency depends on the domain and the attacker strength. On the other hand, we do include in our analysis several protocols devoted to verifiable federated private averaging and verifiable cross-device FL since the same verification techniques could be used in the cross-silo FL setting.

## 2.5   Threat models

In the scope of the considered works, authors usually rely on two widely-spread types of threat models: honest-but-curious (a.k.a. semi-honest) and malicious. According to the standard cryptography definitions, an *honest-but-curious* agent does not deviate from the protocol, but keeps a record of the protocol transcript and analyzes it to gain information about other users, while a *malicious* adversary can deviate from the prescribed protocol instructions and follow an arbitrary strategy to obtain greater benefits. However, in the context of FL, authors often adapt these definitions with additional properties. In order to thoroughly analyze miscellaneous flavors of the applied threat models we distinguish the following four categories:

- **honest (or trusted)**: always follows the protocol correctly and is trusted with sensitive information;

- **honest-but-curious**: always follows the protocol correctly, but is not trusted with sensitive information;

- **forger**: may try to forge different data, but otherwise follows the protocol, is not trusted with sensitive information;

- **malicious**: can arbitrary deviate from the protocol and is not trusted with sensitive information.

In the scope of this paper, in order to describe different approaches in a rigorous manner, we specify the robustness of protocols to participant drop-outs (marked with $\Delta$), i.e. agents who register to participate but subsequently abandon the protocol, separately from the aforementioned categories of threat models. For example, to report that a protocol is robust both against forging and drop-outs, we denote its threat model as "forger $+ \Delta$".

## 2.6   Blockchain technology

A blockchain is a sequence of blocks, which holds a complete list of transaction records like a conventional public ledger (Wang et al., 2018). A transaction could be any action taking place on a blockchain network, for example, a transfer of digital currency from one party to another. In order to achieve the agreement about the ledger's state, blockchains rely on consensus mechanisms. For instance, one of the most popular mechanisms is a Proof of work (PoW), where parties, called miners, calculate a hash of the constantly changing block header. When one miner obtains a relevant value, all others must confirm its correctness and add a collection of transactions used for the calculations as a new block.

In certain blockchain settings, parties can deploy on the blockchain code scripts, called smart contracts, that run synchronously on multiple nodes of the blockchain. Smart contracts can be stored in the blockchain and can be automatically executed when certain pre-conditions are met (Zou et al., 2021).

## 2.7 Commitment scheme

A commitment scheme (CS) is a cryptographic primitive that allows parties to commit to values while keeping them hidden from others (Blum, 1983). A party cannot modify the value after committing to it, but can later reveal it. We define a CS formally as follows (Bünz et al., 2020):

*Definition (Commitment scheme).* A commitment scheme is a triple of algorithms (Setup, Commit, Open) where:

- **Setup**$(1^\lambda) \to$ pp generates public parameters pp;

- **Commit**$(\text{pp}, x) \to (C; r)$ takes a private value $x$ and outputs a public commitment $C$ and (optionally) a secret opening hint $r$;

- **Open**$(\text{pp}, C, x, r) \to b \in \{0, 1\}$ verifies the opening of commitment $C$ to the message $x$ provided with the opening hint $r$.

CSs often have two properties: hiding, meaning that a commitment reveals nothing about the original value, and binding, meaning that a party cannot find other data yielding the same commitment, typically due to a computationally hard underlying problem. Lastly, some CSs are also homomorphic, meaning that there are two binary operations $+$ and $\cdot$ defined in the domain of the original values and the domain of their commitments respectively, such that the following condition holds: $Commit(x + y) = Commit(x) \cdot Commit(y)$, where $x, y$ are values possessed by a party (or parties).

## 2.8 Zero-knowledge proof

Informally, ZKPs are cryptographic methods that enable a party, referred to as the prover, to convince another party, the verifier, of the validity of a given statement without revealing additional information. In FL, the statement is typically represented as the satisfiability of an arithmetic circuit, which encodes computations performed by an untrusted party. For instance, such a circuit can encode the execution of the forward pass of a neural network, which is expected to be calculated by a client. Using a ZKP scheme, a proving party can try to convince a verifying party(ies) that it knows a witness $w$ such that the circuit is satisfied given a public input\output $x$.
In this work, we primarily discuss Non-Interactive Zero-Knowledge (NIZK) proofs (Morais et al., 2019):

*Definition (Non-Interactive Zero Knowledge proof scheme).* A NIZK proof scheme is defined as a triple of algorithms (Setup, Prove, Verify), where:

- **Setup**$(1^\lambda) \to (pk, vk)$ generates parameters: the proving key $pk$ and the verification key $vk$;

- **Prove**$(pk, x, w) \to \pi$ takes as input the instance $x$, the witness $w$, and outputs the zero-knowledge proof $\pi$;

- **Verify**$(vk, x, \pi) \to b \in \{0, 1\}$ receives the proof as input and outputs $b$, which is equal to 1 if the verifier accepts the proof.

Various proving schemes exhibit different properties, such as knowledge-soundness, succinctness, computational/statistical zero-knowledge. In this survey, we use "ZKP" to refer to a general family of proving schemes and "proof" to denote the outcome of the proving algorithm. We invite the reader to (Thaler, 2022) for more details.

# 3 Verifiable cross-silo FL protocols analysis

In this section, we discuss the feasibility of verification in the FL context, present a taxonomy of existing verifiable cross-silo FL protocols, analyze the efficiency of verification techniques and threat models, and discuss the impact of the cross-silo setting on verification. In the scope of this section, we refer to the number of clients as $C$ and to the number of the aggregated vector dimensions as $D$.

## 3.1 Feasibility of verification in privacy-preserving FL

Before presenting the taxonomy, we will consider some more general issues. One could wonder whether it is possible to verify the whole FL process. To answer that question, it is helpful to describe the process of FL in two separate steps:

- First, DOs commit to their data. Approaches that do not consider the threat of forgers don't strictly need this step, as they trust the DOs to use correct data. Several approaches use cryptographic commitment schemes (see Section 2.7) or techniques with similar properties.

- Second, the computations are performed. In this phase, the commitments are used (a) to ensure that values used in the computation are the same values as those committed to, and (b) to ensure that the required relations hold between the (commitment to) inputs and (the observed) output.

In principle, one can prevent poisoning by using verification to show that the inputs the DOs committed to and the outputs satisfy the desired relations. However, it is important to note that except for trivial cases it is not possible to both guarantee privacy, e.g., in the form of differential privacy, and fully prevent data poisoning. Indeed, we can see this as follows. Recall that datasets are called adjacent if they differ in at most one instance. Let $A$ be a $(\epsilon, \delta)$-differentially private algorithm (Dwork & Roth, 2014), i.e., for any pair of adjacent datasets $D$ and $D'$ and for every subset $Y$ of possible outputs, there holds $P(A(D) \subseteq Y) \leq e^\epsilon P(A(D') \subseteq Y) + \delta$, i.e., the distributions of outputs of $A$ for the inputs $D$ and $D'$ are indistinguishable (with tolerance $(\epsilon, \delta)$). Now suppose that the private data of a DO has actually value $x$, and the DO commits to value $\hat{x}$ to be used in the computation. Deciding whether there is data poisoning is equivalent to deciding whether $x = \hat{x}$. However, differential privacy dictates that the distributions of the outputs of $A$ are indistinguishable (up to the tolerance $(\epsilon, \delta)$) if in the input (only) $\hat{x}$ is replaced by $x$, so if $x \neq \hat{x}$ one shouldn't be able to infer this from the output. Hence, requiring differential privacy prevents one from detecting data poisoning (of a single instance in a data set) by (only) observing the output of the private algorithm and the private verification.

For other notions of privacy, similar arguments can be made. In general, enforcing privacy will allow (possibly limited forms) of data poisoning to be undetectable by any verification strategy. For example, when a DO poisons the data before committing to it. In such cases, one could still ask the DO to prove that the committed value $\hat{x}$ is in the proper domain. Moreover, the commitments have the benefit that a DO who uses incorrect inputs will need to stick to the same incorrect values in all other computations using the same commitments. Note that privacy doesn't affect the ability of DOs to commit to incorrect data, a malicious DO has this option both when there is privacy, e.g., differential privacy, and when there is none. One can also resort to other sources of trust, e.g., the commitment to $\hat{x}$ may be the same as an already known commit to a private output of a previous verified computation, one can let a trusted third party verify that $x = \hat{x}$ if the DO can show to this trusted party both $x$, $\hat{x}$ and a proof that $x$ is the real data while trusting that the third party will not disclose the private value.

For model poisoning the situation is different, as this concerns computations after the DOs committed to their data. For example, methods providing differential privacy can be made robust against model poisoning. Even though the output itself is more noisy and hence verification based on the output only is more difficult, approaches such as ZKPs allow one to verify the correctness of the output before adding noise without revealing that sensitive output, and to verify that the noise added afterwards is drawn from the correct distribution (see e.g., Sabater et al. (2023)).

To conclude, while privacy-preserving FL may allow certain forms of data poisoning to remain undetectable by any verification strategy, ensuring privacy does not necessarily hinder verifiability. As discussed above, in many cases verification and privacy can be to a large extent combined.

Lastly, we note that in the literature there exist alternative strategies to cope with other robustness issues next to poisoning attacks, e.g., non necessarily malicious byzantine failures that may cause among others outliers in the input distribution. Such Robust FL approaches often focus on eliminating outliers or mitigating the effects of outliers by using aggregation techniques less sensitive to outliers (Guerraoui et al., 2024). Such approaches are to some extent orthogonal to the verification approaches discussed in the current paper as they are concerned with statistical properties of the data distribution while verifiable computation discussed here starts from data for which the data owner claims it was generated from the right distribution. It is possible to combine verifiable computation and outlier-robust techniques, e.g., one could use a median aggregation rather than an average aggregation to decrease the effect of outliers, and then verify that the median is computed correctly. As stated in Section 2.4, in the current paper we don't consider distribution-level strategies.

## 3.2 Taxonomy description

In order to ensure that a FL protocol is executed correctly, one has to verify both the local computations and the aggregation. We distinguish four categories of different verification techniques and describe each of them in details below. The taxonomy is presented in Figure 1, we also present a timeline of protocols with respect to the taxonomy categories in Figure 2. Although each approach has specific characteristics, our categories allow observing general design patterns and infer conclusions about their efficiency. For this purpose, we assess computational and communication costs both per client and per server for each method. The comparison of asymptotic complexities of protocols and applied threat models is presented in Table 1 for approaches focused on verifiable aggregation and in Table 2 for approaches focused on verification of local computations. We emphasize that complexity metrics are calculated specifically for the verification overhead and do not reflect computation and communication which is needed even if no verification is performed. Lastly, we assume that public key infrastructure, ML model weights, and seeds of PRGs are initialized before the training procedure and do not require a presence of a trusted party.

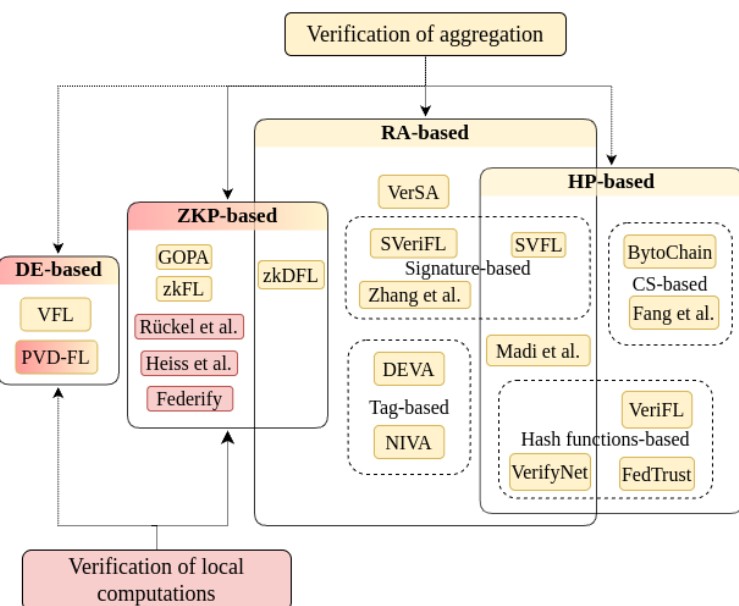

Figure 1: A taxonomy of verifiable cross-silo FL protocols. The red color corresponds to approaches focused on the verification of clients' computations, the yellow color is used for approaches focused on the aggregation verification.

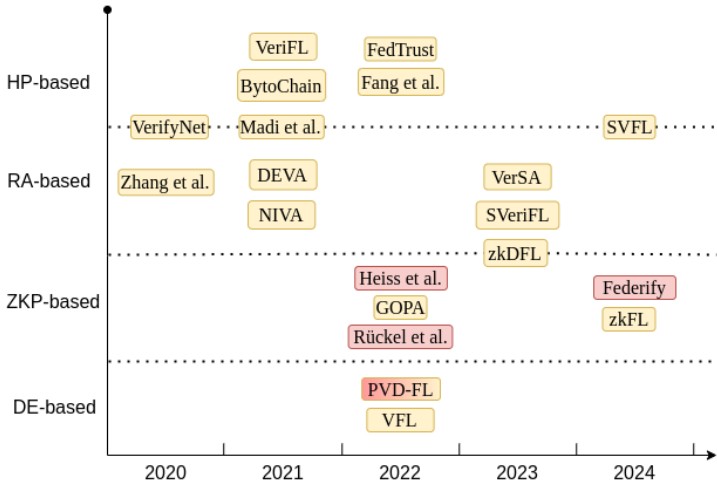

Figure 2: A timeline of verifiable cross-silo FL protocols with respect to the taxonomy categories. The colors are identical to the ones in Figure 1.

In order to fairly compare threat models of different approaches, we mapped the threat models described in the considered papers to the definitions from Section 2.5 and assigned a suitable model ourselves in cases where the authors did not explicitly describe it. In such cases, in tables 1 and 2, we use square brackets to denote assigned threat models and collusion markers. Additionally, some articles provided descriptions of multiple threat models, for example, separate models from privacy and verification perspectives; our table reflects only those related to verification. We also note that assigning a threat model for approaches based on blockchain infrastructure poses additional challenges. While clients' computations in such settings are performed by the clients themselves, aggregation typically occurs as a smart contract. As a result, all miners (or a set of miners, see Section 3.3) redundantly compute the aggregation result. In such cases, in Tables 1 and 2, the server's threat model corresponds to the threat model of miners. However, it is important to mention that the blockchain infrastructure itself is also vulnerable to specific threats, which are often not detailed in articles devoted to blockchain-based FL. As an example, Fang et al. (2022) explicitly mention that the protocol is robust when 70% of the stake in the system is honest, but for others such assumptions are not always easily visible. In tables 1 and 2, the symbol "*" corresponds to a threat model applied to a fraction of agents. To sum up, threat models described in our tables for blockchain-based approaches are meaningful only when the assumptions of blockchain infrastructure itself are satisfied.

In tables 1, 2, as mentioned in 2.5, we mark with "$\Delta$" methods which are robust against client drop-out and/or server drop-out. For non-interactive methods (marked with "$\Delta^\dagger$"), clients only interact once to submit their contribution, which results in a rather trivial robustness against client drop-out: either a client submits the contribution and a later drop-out of the client is not relevant, or the client does not submit the contribution and then doesn't participate at all. For such approaches, once a client participates, his contribution will be taken into account in the computed aggregate if the protocol finishes. For interactive protocols, in all cases clients need to contribute to some form of "decryption" of the aggregate. Some approaches (Sabater et al., 2022) try to remove the input of the client(s) who dropped out so the set of clients who decrypt is exactly the set of clients who contributed input. In other approaches (Xu et al., 2020; Hahn et al., 2023; Tsaloli et al., 2021), robustness against client drop-out is achieved by a threshold secret sharing where only a fraction of clients is needed for this decryption. This decreases security, but avoids the need for computations to be rolled back. When we mark the server threat model as robust against drop-out, in all cases there are multiple servers (in the case of blockchain based methods multiple miners) of which a certain fraction can drop out without preventing the computation of the final result, e.g., due to a threshold secret sharing scheme among servers or due to the consensus properties of blockchains.

**Redundant aggregation (RA) based verification.** This category consists of approaches that require the server to aggregate some redundant values in order to prove that the aggregation of clients' values is performed

Table 1: A comparison of asymptotic complexities of the aggregation verification overhead per epoch and threat models of verifiable FL protocols. Sections correspond to taxonomy categories (Figure 1). Notations: $E$ – number of epochs, $C$ – number of clients, $D$ – number of vector dimensions, BC – requires a blockchain infrastructure (the (significant) cost of the blockchain is not included in the table), S-C col. – Server-Client collusion to bypass the verification, [. . .] – not explicitly described in the paper.

| Approach | Computational cost | | Communication cost | | Threat model | | S-C col. | TA | BC |
|---|---|---|---|---|---|---|---|---|---|
| | client | server | client | server | client | server | | | |
| VerSA | $O(D)$ | $O(CD)$ | $O(D)$ | $O(CD)$ | h-b-c + Δ | [forger] | ✗ | ✗ | ✗ |
| SVeriFL | $O(D)$ | $O(CD)$ | $O(D)$ | $O(CD)$ | h-b-c | forger | ✗ | ✓ | ✗ |
| Zhang et al. | $O(D)$ | $O(C)$ | $O(1)$ | $O(C)$ | [hon] | forger | [✗] | ✗ | ✗ |
| DEVA | $O(CD)$ | $O(CD)$ | $O(CD)$ | $O(CD)$ | h-b-c + Δ | forger + Δ | ✗ | ✗ | ✗ |
| NIVA | $O(CD)$ | $O(CD)$ | $O(CD)$ | $O(CD)$ | [h-b-c + Δ†] | [forger + Δ] | [✗] | ✗ | ✗ |
| SVFL | $O(D)$ | $O(C)$ | $O(1)$ | $O(C)$ | h-b-c | [forger] | ✗ | ✗ | ✗ |
| Madi et al. | $O(D)$ | $O(C)$ | $O(1)$ | $O(D)$ | hon | [forger] | ✗ | ✗ | ✗ |
| VerifyNet | $O(D)$ | $O(CD)$ | $O(D)$ | $O(CD)$ | h-b-c + Δ | forger | ✗ | ✗ | ✗ |
| BytoChain | $O(C+D)$ | $O(D)$ | $O(1)$ | $O(1)$ | [h-b-c + Δ†] | [forger* + Δ] | [✗] | ✗ | ✓ |
| Fang et al. | $O(C+D)$ | $O(1)$ | $O(1)$ | $O(1)$ | [hon + Δ†] | [forger* + Δ] | [✗] | ✗ | ✓ |
| VeriFL | $O(C + \frac{D}{E})$ | $O(1)$ | $O(C)$ | $O(1)$ | h-b-c + Δ | forger | [✗] | CRS | ✗ |
| FedTrust | $O(CD)$ | $O(1)$ | $O(CD)$ | $O(1)$ | hon | forger + Δ | [✗] | ✗ | ✗ |
| zkDFL | $O(D)$ | $O(CD)$ | $O(C)$ | $O(C)$ | [hon + Δ†] | [forger*+Δ] | [✗] | CRS | ✓ |
| GOPA | $O(D\log C)$ | N/A | $O(D\log C)$ | N/A | forger* + Δ | N/A | N/A | ✗ | ✗ |
| zkFL | $O(CD)$ | $O(CD)$ | $O(1)$ | $O(C\log(CD))$ | [hon + Δ†] | [forger+Δ] | [✗] | ✗ | ✗/✓ |
| VFL | $O(D)$ | $O(1)$ | $O(1)$ | $O(1)$ | h-b-c | [forger] | [✗] | ✗ | ✗ |
| PVD-FL | $O(D)$ | N/A | $O(D)$ | N/A | [forger*] | N/A | N/A | ✗ | ✗ |

Table 2: A comparison of asymptotic complexities of the local computations verification overhead per epoch and threat models of verifiable FL protocols. Sections correspond to taxonomy categories (Figure 1). The notations are identical to the ones in Table 1.

| Approach | Computational cost | | Communication cost | | Threat model | | S-C col. | TA | BC |
|---|---|---|---|---|---|---|---|---|---|
| | client | server | client | server | client | server | | | |
| Rückel et al. | O(D) | O(1) | O(1) | O(1) | [forger + Δ†] | [forger* + Δ] | ✗ | CRS | ✓ |
| Heiss et al. | O(D) | O(1) | O(1) | O(1) | [forger + Δ†] | [forger* + Δ] | ✗ | CRS | ✓ |
| Federify | O(D) | O(1) | O(1) | O(1) | [forger + Δ†] | [forger* + Δ] | ✗ | CRS | ✓ |
| PVD-FL | $O(D)$ | N/A | $O(D)$ | N/A | [forger*] | N/A | N/A | ✗ | ✗ |

---

**Algorithm 1** Generic RA-based verification

**For a client $\mathcal{C}_i$ :**
  **Input:** Private data $d_i$
  1. Receive the global model $\theta$ from the server
  2. Compute the local model $\theta_i$ using $d_i$ and $\theta$
  3. Compute the ciphertext/tag $t_i$ using $d_i$ and/or $\theta_i$
  4. Send $\theta_i$ and $t_i$ to the server
  5. Receive the updated global model $\theta^+$ and the aggregated ciphertext/tag $t$ from the server
  6. Verify the correctness of $\theta^+$ using $t$

**For a server $\mathcal{S}$ :**
  **Input:** Global model $\theta$
  1. Send the global model $\theta$ to each client
  2. Receive ciphertexts/tags $\{t_i\}_{i=1}^{C}$ and local models $\{\theta_i\}_{i=1}^{C}$ from all clients
  3. Compute the updated global model $\theta^+$ by aggregating $\{\theta_i\}_{i=1}^{C}$
  4. Compute the aggregated ciphertext/tag $t$ by aggregating $\{t_i\}_{i=1}^{C}$
  5. Send $\theta^+$ and $t$ to all clients

correctly. This feature leads to a computational cost of the server to be at least $O(C)$. Moreover, some protocols (Tsaloli et al., 2021; Brunetta et al., 2021; Hahn et al., 2023) are designed under the assumption that each party has one secret value, therefore a naive scaling of the approach to a FL setting where parties share multi dimensional data would lead to an additional factor $D$ in the complexity. We present a generic protocol with a RA-based verification of aggregation in the Algorithm 1.

In (Zhang et al., 2020; Gao et al., 2023) authors proposed to check that the result of aggregation is correct by means of cryptographic signature schemes based on bilinear pairings. In both works, the server has to compute a redundant aggregation of signatures. Indeed, such schemes allow ensuring that the aggregation result is obtained from data signed by all other clients, however a malicious server may aggregate arbitrary signed values (e.g., clients' values from previous epochs) and successfully pass the verification with a fabricated aggregated value. As a result, by relaxing the threat model, (Zhang et al., 2020) achieves better complexity than other protocols of the same category.

In DEVA (Tsaloli et al., 2021) and NIVA (Brunetta et al., 2021) decentralized FL protocols are proposed, all clients compute tags corresponding to the secret clients' values which are later used to verify that the aggregation is correct. Both protocols have an additional $D$ complexity factor due to the assumption that each client has only one secret value. In contrast to many others, DEVA and NIVA present solutions for a setting with multiple servers. Another RA-based approach is VerSA (Hahn et al., 2023). In VerSA all clients use the same secret vectors to compute a new input from their secret values and later check the consistency of the aggregated result with the aggregation of new inputs. Interestingly, the authors provide a comparison with VerifyNet (Xu et al., 2020) approach (which lies at the intersection of RA- and HP-based categories) showing that despite the same verification asymptotic complexity, the VerSA's cost is orders of magnitude smaller.

Finally, from the treat model perspective, all RA-based approaches cope with a forger server while considering clients to be honest or honest-but-curious. Additionally, in SVeriFL (Gao et al., 2023), the integrity of the data shared by clients is verified, this threat is processed by addition of a trusted authority (TA).

**Homomorphic property (HP) based verification.** This category covers verification techniques which rely on the HP of underlying primitives: hash functions (Guo et al., 2021; Hsu et al., 2022) and commitment schemes (Li et al., 2021; Fang et al., 2022). The general idea of such protocols is the following: clients compute hashes/commitments from their data and share them with each other, then all clients may verify the result of aggregation by checking that this result corresponds to the aggregation of hashes/commitments through homomorphism. As a consequence, both computational and communication costs of the server are $O(1)$ if the ciphertext length does not depend on $C$ or $D$. In order to verify the aggregation of clients' values, clients have to compute a hash/commitment in $O(D)$ from their data and to aggregate hashes/commitments from other clients in $O(C)$. In blockchain-based protocols, clients upload hashes/commitments with a constant communication cost (to be multiplied with costs related to the blockchain infrastructure, see Section 3.3), while in other protocols, where clients have to exchange messages with each other, the communication cost per client is at least $O(C)$. Exceptionally, in FedTrust (Hsu et al., 2022) client costs have an additional complexity factor $D$, as far as the hash is calculated for each component separately. We present a generic protocol with a HP-based verification of aggregation in the Algorithm 2.

In VeriFL (Guo et al., 2021), apart from the main protocol, authors also demonstrated a way to optimize the verification. Leveraging the homomorphic property of the hash scheme and the repetitive nature of FL calculations, authors proposed an amortized verification which allows decreasing the number of hash function calls. Instead of checking that the product of hashes obtained from all clients is equal to the hash of the aggregated value obtained from the server after each epoch, one can draw a set of random coefficients to compute a linear combination of hashes obtained from all clients across multiple epochs and compare it with a hash of a linear combination of aggregated values obtained from the server using the same coefficients. As a result, in Table 1, the client computational cost of VeriFL differs from competing protocols by additional $\frac{1}{E}$ factor. Notable, since the core idea of all approaches of HP-based category is very similar, we observe that this optimization could be also applied to any of them.

There are also few protocols that lie at the intersection of the RA- and HP-based categories: VerifyNet (Xu et al., 2020), Madi et al. (2021) and SVFL (Luo et al., 2024). In VerifyNet, clients share additional values

---

**Algorithm 2** Generic HP-based verification

---

**For a client** $\mathcal{C}_i$ :
  **Input:** Private data $d_i$
  1. Receive the global model $\theta$
  2. Compute the local model $\theta_i$ using $d_i$ and $\theta$
  3. Compute the hash $h_i$ of $\theta_i$
  4. Send $\theta_i$ to the server
  5. Publish $h_i$
  6. Receive the updated global model $\theta^+$
  7. Collect hashes from other clients $\{h_j\}_{j=1, j \neq i}^{C}$
  8. Check whether the hash of $\theta^+$ equals to the aggregation of hashes $\{h_j\}_{j=1}^{C}$

**For a server** $\mathcal{S}$ :
  **Input:** Global model $\theta$
  1. Send the global model $\theta$ to each client
  2. Receive $\{\theta_i\}_{i=1}^{C}$ from all clients
  3. Compute the updated global model $\theta^+$ by aggregating $\{\theta_i\}_{i=1}^{C}$
  4. Send $\theta^+$ to all clients

---

along with their gradients, which are later aggregated by the server. Aggregated values are shared back with clients so that they can verify the correctness of the gradients aggregation relying on the properties of the homomorphic hash function. As a result, although the computational complexity of the protocol follows the trend of the RA-based category, the underlying verification technique shares similarities with those employed in HP-based protocols. Similarly, SVFL is based on a homomorphic signature scheme. During the training procedure, the server performs a redundant aggregation of signatures and each client runs the verification algorithm based on the HP. Note that Luo et al. (2024) considered a malicious threat model for the server, however one should be careful while using signature-based verification techniques due to the concerns described above for similar approaches of the RA-based verification category (Gao et al., 2023; Zhang et al., 2020).

**ZKP based verification.** The third category contains approaches which are based on ZKPs. The core principle could be described as follows: a party performs calculations and at the same time computes the proof, which is shared along with the result of calculations; other parties can later run the proof verification algorithm to ensure that the result was computed correctly. In contrast to previous categories, ZKPs allow proving arbitrary computations, therefore such methods are suitable for proving the correctness of both aggregation of clients' values (Wang et al., 2024; Ahmadi & Nourmohammadi, 2023; Sabater et al., 2022) and local computations (Keshavarzkalhori et al., 2024; Heiss et al., 2022; Rückel et al., 2022). Moreover, advanced ZKP schemes provide a proof size that is sublinear in the number of computations to prove. We present a generic protocol with a ZKP-based verification of local computations in Algorithm 3.

In protocols focused on aggregation, authors build on different infrastructures and ZKP schemes. In GOPA (Sabater et al., 2022), authors introduced a decentralized gossip approach where nodes publish proofs of their computations using $\Sigma$-protocols (Attema & Cramer, 2020). In zkFL (Wang et al., 2024) the authors apply a more modern ZKP scheme, Halo2 (Bowe et al., 2019), and provide two versions of the protocol: with a centralized FL setting and a blockchain based one. In zkDFL (Ahmadi & Nourmohammadi, 2023) authors rely on the blockchain infrastructure and the Groth16 scheme (Groth, 2016). We also observed that the authors of zkDFL and zkFL use different techniques to prove that each client value was indeed sent by one of the clients. In zkFL, all clients sign commitments to their local models, and the server includes the signature verification in the proof of own computations. Interestingly, with this example one can notice that ZKPs help to eliminate the need for clients to communicate with each other to verify the integrity of aggregated data. In zkDFL, authors proposed to deploy a smart contract which checks that the result of redundant aggregation of local weights hashes performed by server is equal to the sum of hashes uploaded by clients (notably, computational burden of the aggregation verification is also transferred to the smart

---

**Algorithm 3** Generic ZKP-based verification

---

**For a client** $\mathcal{C}_i$ :
  **Input:** Private data $d_i$
  1. Receive the global model $\theta$
  2. Compute the local model $\theta_i$ using $d_i$ and $\theta$
  3. Run NIZK.Prove to compute the proof $\pi_i$ using intermediate values of the step 2 as a witness
  4. Send the local model $\theta_i$ to the server
  5. Receive the updated global model $\theta^+$

**For a server** $\mathcal{S}$ :
  **Input:** Global model $\theta$
  1. Send the global model $\theta$ to each client
  2. Receive local models $\{\theta_i\}_{i=1}^C$ and proofs $\{\pi_i\}_{i=1}^C$ from all clients
  3. Run NIZK.Verify for each $\pi_i$ to verify the correctness of local models
  4. Compute the updated global model $\theta^+$ by aggregating $\{\theta_i\}_{i=1}^C$
  5. Send $\theta^+$ to all clients

---

contract). Consequently, differences in the settings and chosen ZKP schemes result in different complexity metrics.

In contrast to protocols with a verifiable aggregation, approaches focused on local computations verification (Rückel et al., 2022; Heiss et al., 2022; Keshavarzkalhori et al., 2024) have almost identical design: all protocols use the Groth16 scheme to achieve verifiability of clients' local computations in a blockchain based setting. However, authors focus on verification of different ML models: a linear regression model in (Rückel et al., 2022), a naive Bayes classifier in (Keshavarzkalhori et al., 2024) and a feedforward neural network in (Heiss et al., 2022). In all three approaches, aggregation is performed by a smart contract, therefore the correctness of the aggregation relies on the blockchain security assumptions. One can notice that proving, verification, and proof size complexities would be the same for the same type of computations, since all three protocols use the same ZKP scheme. Similarly, in all three protocols participants need to have an access to public parameters (including a Common Reference String (CRS)) which would also require the same amount of memory for storage.

Additionally, in (Rückel et al., 2022) clients commit to the Merkle root of their private dataset while registering in the system which allows ensuring data integrity within the whole training procedure. To achieve this, clients extend each proof showing that computations are performed with committed data. In Federify (Keshavarzkalhori et al., 2024), ZKPs are also applied to prove properties of local models, i.e., distance metrics indicating how close the submitted model is to the global one. As a result, both approaches prevent the risk of data poisoning to the extent possible according to Section 3.1.

Lastly, it is important to mention that there are many research works devoted to proving the correctness of ML calculations without a FL setting (Huang et al., 2022b; Zhao et al., 2021; Hu et al., 2023). Often, the same verification techniques and developed optimizations could be applied in the context of FL to prove local computations of participants. We believe that the vast majority of such techniques would fit the ZKP-based verification category of our taxonomy.

**Data Embedding based verification.** This category covers protocols, where participants embed additional values into their data before sharing it with untrusted parties; later, the result of calculations performed by an untrusted source is assumed to be correct if the corresponding additional values are computed correctly. The embedding principle leads to an increase in the size of the transmitted data and the complexity of the outsourced calculations, which depends on the size of embedded values, and requires more expensive data preprocessing. We present a generic protocol with DE-based verification of aggregation in the Algorithm 4.

One of the best approaches focused on the verifiable aggregation from the complexity perspective is VFL (Fu et al., 2022). In VFL clients encode their secret values as a polynomial function and embed an additional point $(a_i, A)$ before interpolation, where $a_i$ is a parameter of client $i$ and $A$ is obtained from a pseudo

---

**Algorithm 4** Generic DE-based verification

---

**For a client $\mathcal{C}_i$ :**
  **Input:** Private data $d_i$
  1. Receive the global model $\theta$
  2. Compute the local model $\theta_i$ using $d_i$ and $\theta$
  3. Calculate the modified local model $\hat{\theta}_i$ by embedding additional values $a$
  4. Send the modified local model $\hat{\theta}_i$ to the server
  5. Receive an updated global model $\hat{\theta}^+$
  6. Verify the correctness of $\hat{\theta}^+$ using values $a$

**For a server $\mathcal{S}$ :**
  **Input:** Global model $\theta$
  1. Send the global model $\theta$ to each client
  2. Receive $\{\hat{\theta}_i\}_{i=1}^C$ from all clients
  3. Compute the updated global model $\theta^+$ by aggregating $\{\hat{\theta}_i\}_{i=1}^C$
  4. Send $\hat{\theta}^+$ to all clients

---

random generator (PRG) using $a_i$ as input. In order to verify the result of aggregation performed by a malicious server, each client checks that the evaluation of the aggregated polynomial function at the point $a_i$ corresponds to the output of PRG with $\sum_{i=1}^C a_i$ as an input. In this protocol, transmitted data overhead is negligible, but each client has to compute a costly interpolation and stores large public parameters.

In PVD-FL (Zhao et al., 2022), authors proposed a decentralized verifiable protocol which is based on the verifiable matrix multiplication algorithm. Parties embed random vectors into their data and then check that these vectors were correctly multiplied, as a result, the developed algorithm allows to verify basic operations of ML. In contrast to other approaches surveyed in this paper, in PVD-FL authors aim to show correctness of both aggregation and local models calculation. However, they also mention that their protocol is still vulnerable to poisoning attacks.

### 3.3 Discussion

In this subsection, we discuss how various features of the cross-silo setting impact the development of a verifiable FL protocol. We highlight efficient schemes which cope with cross-silo FL challenges, describe the influence of the threat model choice on the efficiency and security of the protocol, and discuss advantages and disadvantages of the blockchain-based approaches.

Firstly, we describe the link between cross-silo setting characteristics and the efficiency of protocols. One can notice that in tables 1 and 2 there are mainly two parameters determining the communication cost: $C$ and $D$. However, there is a large difference in their impact on complexity. Since the number of participants in the cross-silo settings is moderate while ML models typically have large sizes, a dependence on $D$ is less desirable. Nevertheless, taking into account that clients anyway must send their local models to a server with $O(D)$ communication cost, the overall FL complexity would become asymptotically worse only in cases when the verification overhead is larger than $D$. For example, such as in (Tsaloli et al., 2021; Brunetta et al., 2021; Hsu et al., 2022), where the communication cost is $O(CD)$.

Secondly, we observe that several approaches rely on a blockchain infrastructure (Fang et al., 2022; Li et al., 2021; Ahmadi & Nourmohammadi, 2023; Rückel et al., 2022; Heiss et al., 2022). This strategy offers several advantages. For instance, smart contracts enforce a transparent and verifiable distribution of incentives (Rückel et al., 2022). The use of smart contracts to perform aggregation also makes the presence of a distinct server unnecessary, thereby replacing a single party trust with blockchain trust guarantees. All blockchain-based approaches are also robust against limited drop-outs of aggregators, i.e. miners. Table 1 demonstrates that the verification overhead per client is generally smaller for blockchain-based approaches compared to non-blockchain protocols with similar verification techniques; however, there is a significant infrastructure overhead, e.g., costs of the miners, which is a critical limitation within the context of cross-silo

FL. The miners have to execute identical calculations, resulting in a tremendous total computational burden across all participants. We note that the miners' overhead can vary across different blockchains depending on the consensus mechanism: while in Proof-of-Work (PoW) systems like Bitcoin, all miners redundantly execute all computations, in Proof-of-Stake (PoS) systems like Ethereum, transactions are verified by a set of parties called "validators". Although the general idea remains the same, the set of validators is smaller, reducing the overall computational cost. Moreover, blockchain-based approaches suffer from latency issues: the time required to confirm each block can slow down the FL training. Since FL computations are typically non-parallelizable, for example, due to the sequential nature of neural network training, and may contain thousands of iterations, latency poses a significant barrier to real-world deployment. Another challenge is the intensive energy consumption associated with costly consensus mechanisms. We invite the reader to Wenhua et al. (2023) for more details on the challenges related to blockchains. Finally, in cross-silo FL there is typically at least one party interested in obtaining the results of training, thus there is no a strong need for a decentralized infrastructure.

Thirdly, during our analysis of verifiable aggregation protocols, we observed that server has several ways to fabricate the aggregation result, and sometimes a proposed protocol only addresses a portion of them. Specifically, there are three ways in which a forger can manipulate the aggregation of clients' values: omitting one of the values, replacing one update with arbitrary data or inserting an additional update. However, in RA-based approaches that rely on cryptographic signature schemes, a verifier is only capable of checking that server has not omitted values from other clients and has not inserted additional values in the sum. Nevertheless, a malicious server still may aggregate arbitrary signed values and successfully pass the verification. Therefore, such verification techniques should be employed with caution.

Lastly, our observations indicate that the majority of verifiable aggregation protocols rely on either redundant aggregation mechanisms or leverage the homomoprhic properties of cryptographic primitives. However, these techniques are not suitable for proving complex computations, such as those encountered in ML. To address this limitation, researchers employ ZKP schemes, known for the ability to prove arbitrary computations. We examined several ZKP-based protocols developed both to verify aggregation and more complex computation of clients. Furthermore, ZKP schemes offer the flexibility to reinforce the primary proof with additional information, showing the correctness of computed noise, data provenance, or distance metrics. The complexity metrics of ZKP-based FL protocols directly depend on the chosen scheme, however we observe the lack of justifications behind the scheme selection process in the literature. Considering the benefits of ZKPs and their advantage in comparison with other techniques, in the next section we examine existing ZKP schemes with a focus on their applicability in the cross-silo setting.

## 4 ZKP for cross-silo FL

The development of new ZKP schemes has been an active area of research over the past decade. While the surge of new protocols has led to a broad variety of schemes to choose from, it also resulted in additional desirable characteristics, making it challenging to determine the most suitable choice for a specific application. In this context, we discuss the applicability of ZKPs in cross-silo FL and study how to prove calculations in this setting minimizing the cost.

### 4.1 Applicability

In this subsection, we discuss diverse characteristics of ZKP schemes, examining their implications within the context of the cross-silo FL setting. Specifically, we consider the time complexities for proving, verifying and preprocessing, the proof size, the CRS and the commit-and-prove property which some ZKP schemes feature, and the partition into transparent schemes versus trusted setup based schemes.

**Computational complexity of proving and verifying.** In cross-silo FL, the verification of proofs generated during the training procedure presents a significant challenge: each party who wants to ensure the correctness of the protocol needs to execute a verification algorithm in order to check proofs coming from many participants. If the verification algorithm has a linear cost with respect to the size of the computations, the cost for the verifier is proportional to the combined computations of all other parties in the system, which

is often too expensive. Furthermore, in scenarios where the protocol is publicly verifiable, e.g. when all proofs are stored on a bulletin board as demonstrated in (Sabater et al., 2022; Brunetta et al., 2021), verification of all proofs performed by an external party after the training procedure would become excessively time-consuming. Hence, we consider it is desirable that the verification time complexity is at most logarithmic in the total amount of computation to ensure feasibility within the FL setting. In contrast to the verification complexity, the proving complexity is of slightly less priority since each party has to prove only its own computations once. Consequently, the overhead incurred by executing an algorithm to construct a ZKP is often feasible even if this algorithm is somewhat more expensive.

**Preprocessing computational complexity.** Achieving fast verification and small proofs often comes with a costly preprocessing phase, when the CRS is generated. Since this phase is completed only once before the training process, while the preprocessing complexity is a factor to consider, its impact is generally less critical than for algorithms that are executed within the training loop.

**Proof size.** In a verifiable FL protocol, all parties compute proofs of their local calculations. If the proof size scales linearly with the size of computations, parties need to share data proportional to that computation. Such proofs would require gigabytes to petabytes of storage. In the context of publicly verifiable protocols, this problem becomes even more acute. Therefore, similarly to the requirement for verification cost, it is essential for the proof size to be at most logarithmic.

**Common Reference Strings (CRS).** Numerous NIZK schemes rely on generating a CRS during the setup phase. There exist two types of CRS: universal, capable of supporting all circuits of bounded size, and custom, applicable only for specific circuits. The key advantage of a universal CRS is that it eliminates the need for a per-computation preprocessing Kosba et al. (2020), allowing the same CRS to be reused multiple times for different ML models. However, given that in FL all parties agree on the model to train, i.e. computations to prove, there is no strict need to use a scheme with a universal reference string.

In the context of FL, all participants have to access the CRS to compute and verify proofs. The cost of delivering the CRS is determined both by the size of this CRS, and the challenge to broadcast this CRS in a way such that all recipients can trust it. One should take into account that the complexity of the CRS's size differs depending on the ZKP scheme. Therefore, if the CRS scales linearly to the size of computations to prove and some heavy computations have to be proven, the storage cost will likely become prohibitively large.

**Transparent and trusted setup based setups.** A huge fraction of ZKP schemes depend on trusted setups to generate public parameters. Modern protocols often require only one honest party for this procedure. Since many cross-silo FL protocols are developed under the assumption that at least one honest party is present in the setting, applying such schemes would fit the threat model of these protocols. At the same time, there exist transparent ZKP schemes that eliminate the need for a trusted setup. As a result, integrating such schemes into FL protocol would allow reliance on a more robust threat model.

**Commit-and-prove.** Some ZKP schemes maintain the commit-and-prove property, meaning that the prover can commit to certain data and later prove computations involving this data. As a result, a verifier will be able to check that data has not changed across multiple proofs. In the context of FL, this property is useful to demonstrate that DOs have not changed data during the training. Such a procedure mitigates the risk of one-shot data poisoning attacks and narrows capabilities of potential attackers (see Section 3.1).

Following the discussion above, in Table 3, we present an asymptotic comparison of various ZKP schemes with at most logarithmic size proofs realtive to the size of computations. Our comparison shows that while numerous schemes offer small proof sizes and fast verification, only DARK-based approaches (e.g., SuperSonic, Dew) maintain a constant size of public parameters at the same time.

In general, a verification technique for a FL protocol can be selected as follows. First, the list of assumptions should be specified, e.g., the threat models of all FL participants, the presence of trusted parties, or the presence of a blockchain infrastructure. Secondly, using Tables 1 and 2, one can mark relevant approaches that have been developed under the assumptions listed at the first step. If no approaches fit, one can check whether a combination of several approaches could be suitable. Finally, one should evaluate whether the

Table 3: Asymptotic comparison of ZKP schemes with logarithmic and constant proof size complexity. $C$ is the computation expressed as a circuit, $|C|$ is the number of gates in the circuit, $|N|$ is the length of inputs.

| Scheme | Parameters size | Proving | Verification | Proof Size |
|---|---|---|---|---|
| Dory (Lee, 2021) | $O(|C|)$ | $O(|C|)$ | $O(\log|C|)$ | $O(\log|C|)$ |
| Gemini (space-efficient) (Bootle et al., 2022) | $O(|C|)$ | $O(|C|\log^2|C|)$ | $O(\log|C|)$ | $O(\log|C|)$ |
| Gemini (time-efficient) (Bootle et al., 2022) | $O(|C|)$ | $O(|C|)$ | $O(\log|C|)$ | $O(\log|C|)$ |
| SuperSonic (Bünz et al., 2020) | $O(1)$ | $O(|C|\log|C|)$ | $O(\log|C|)$ | $O(\log|C|)$ |
| DARK-fix (Arun et al., 2023) | $O(1)$ | $O(|C|\log|C|)$ | $O(\log|C|)$ | $O(\log|C|)$ |
| BCCGP (Bootle et al., 2016) | $O(|C|)$ | $O(|C|)$ | $O(|C|)$ | $O(\log|C|)$ |
| Bulletproofs (Bunz et al., 2018) | $O(|C|)$ | $O(|C|)$ | $O(|C|)$ | $O(\log|C|)$ |
| Compressed $\Sigma$-protocol (Attema & Cramer, 2020) | $O(|C|)$ | $O(|C|)$ | $O(|N|)$ | $O(\log(|C|))$ |
| Groth16 (Groth, 2016) | $O(|C|)$ | $O(|C|\log|C|)$ | $O(|N|)$ | $O(1)$ |
| Sonic (Maller et al., 2019) | $O(|C|)$ | $O(|C|\log|C|)$ | $O(N)$ | $O(1)$ |
| GGPR (Gennaro et al., 2013) | $O(|C|)$ | $O(|C|\log|C|)$ | $O(|N|)$ | $O(1)$ |
| Pinochio (Parno et al., 2013) | $O(|C|)$ | $O(|C|\log|C|)$ | $O(|N|)$ | $O(1)$ |
| PLONK (Gabizon et al., 2019) | $O(|C|)$ | $O(|C|\log|C|)$ | $O(|N|)$ | $O(1)$ |
| vnTinyRAM (Ben-Sasson et al., 2014) | $O(|C|\log|C|)$ | $O(|C|\log^2|C|)$ | $O(|N|)$ | $O(1)$ |
| Mirage (Kosba et al., 2020) | $O(|C|)$ | $O(|C|\log|C|)$ | $O(|N|)$ | $O(1)$ |
| Behemoth (Seres & Burcsi, 2023) | $O(|C|)$ | $O(|C|^3\log|C|)$ | $O(|N|)$ | $O(1)$ |
| Dew (Arun et al., 2023) | $O(1)$ | $O(|C|^2)$ | $O(\log|C|)$ | $O(1)$ |

marked approaches meet the complexity requirements of the particular application. In the case of ZKP-based approaches, it is also important to decide on the first priority: fast verification and compact proofs (which typically come together), compact storage for the prover/verifier, or fast proving. Then, one can choose a specific scheme using Table 3. There are also other works, e.g., see (Labs, 2023), that provide the results of an experimental analysis of various schemes, which can help to assess concrete performance costs on top of the asymptotic characteristics. In practice, one might also consider other criteria, such as the availability of an open-source implementation for a specific scheme, the programming language of the code, interface options, or the presence of auditor's reports. For instance, one of the most popular ZKP schemes, Groth16 (Groth, 2016), has many open-source implementations, making it easier to find a suitable one for protocols that prioritize fast verification and compact proofs. We invite the reader to (Liang et al., 2025) for more guidelines on selecting ZKPs in different real-world scenarios.

## 4.2 Storage cost optimization

In the previous subsection, we highlighted that one of the more important criteria for designing a verifiable FL protocol is the total communication cost of ZKP schemes and the associated cost to store the proofs. If parties perform more computations than they want to include in a single proof, they can distribute their computations over multiple proofs. In this subsection we first study what is the best granularity of the proofs under different ZKP schemes and then discuss an alternative idea to prove computations based on recursive proof composition.

As in a ML algorithm the same operations are often repeated many times, e.g., for different data or for different epochs in iterative algorithms, we assume that one can produce a ZKP for all computations by repeatedly proving correct evaluation of a single circuit. Then, we introduce a parameter $k$ representing how many evaluations of this circuit are included per ZKP. In order to understand the effect of grouping less or more computations together in a single proof, we define a function to compute the storage cost as a function of $k$ and then find the optimal number $k$ of grouped circuits to prove, i.e., we find the $k$ with minimum function value.

To begin with, we introduce the following notations:

- $c \in \mathbb{N}$ – the size of the minimal circuit in bits, i.e., the size of the smallest circuit so that the complete algorithm can be represented as a repetition of that circuit;

- $n \in \mathbb{N}$ – the number of circuits to prove;

- $1 \leq k \in \mathbb{N} \leq n$ – the number of circuits grouped for one proof;

- $\delta : \mathbb{N}^3 \to \mathbb{N}$ – the function that, for a certain ZKP scheme, gives the total communication cost $\delta(k; c, n)$ in bits to prove an FL algorithm consisting of $n$ evaluations of a circuit of size $c$, where proofs are given for groups of $k$ circuits;

- $\lambda : \mathbb{N} \to \mathbb{N}$ – the function that returns the proof size $\lambda(c)$ required to prove a circuit of size $c$ in bits for a certain ZKP scheme;

- $\psi : \mathbb{N} \to \mathbb{N}$ – the function that gives the size of the public parameters $\psi(c)$ required to prove circuits of size $c$ in bits for a certain ZKP scheme.

We also note that the cost per bit for the distribution and storage of proofs may be different than the cost per bit for the generation and distribution of public parameters, hence we scale $\psi$ and $\lambda$ with constants $a$ and $b$ to reflect the appropriate weights of a certain ZKP scheme.

Then, for an arbitrary ZKP scheme we can define $\delta$ in the following way:

$$\delta(k; c, n) = a\psi(kc) + \frac{nb\lambda(kc)}{k} \tag{1}$$

One can see that if $k = 1$, then the storage cost is as in a classic verifiable FL protocol without the storage optimization.

One can notice that in Table 3 there are ZKP schemes with the same asymptotic complexities of proof and parameters size. Since in the scope of this optimization, such schemes have the same optimal value of $k$, we group schemes with the same complexities together and describe each group below.

**Group 1: Constant parameters size and constant proof size.**

We define the function $\delta_1$ by instantiating $\psi$ and $\lambda$ with corresponding asymptotic complexities of the Group 1 in the definition of $\delta$:

$$\delta_1(k; c, n) = a + \frac{nb}{k}$$

One can see that in the simple case when a ZKP scheme (for example, the Dew scheme (Arun et al., 2023)) has constant parameters and proof size, the function tends to a minimum with the increase of $k$, thus the optimal $k = n$.

**Group 2: Constant parameters size and logarithmic proof size.**

Several approaches of this group can be found in Table 3: SuperSonic (Bünz et al., 2020) and its improved version called DARK-fix (Arun et al., 2023). In order to find an optimal $k$ for this group, we define $\delta_2$ similarly to the $\delta_1$ using the logarithmic proof size complexity:

$$\delta_2(k; c, n) = a + \frac{nb \cdot \log(ck)}{k}$$

Since for k, c > 1 the function is decreasing, the minimum is also reached at $k = n$.

**Group 3: Linear parameters size and constant proof size.** In table 3, there are many approaches with characteristics of the Group 3: Groth16 (Groth, 2016), Sonic (Maller et al., 2019), Pinochio (Parno et al., 2013), GGPR (Gennaro et al., 2013), PLONK (Gabizon et al., 2019), Mirage (Kosba et al., 2020), Behemoth (Seres & Burcsi, 2023). We define $\delta_3$ as follows:

$$\delta_3(k; c, n) = akc + \frac{nb}{k}$$

We can approximately find the minimum of this function on integers by taking the derivative of the corresponding function on the reals:

$$\frac{d\delta_3(k; c, n)}{dk} = ac - \frac{nb}{k^2}$$

The minimum value of $\delta_3$ is $k_e = \sqrt{\frac{nb}{ac}}$. Since $k_e$ could be less than one, the optimal $k$ is $\min(\max(k_e, 1), n)$. Note, that in practice one has to check which of the two values $\lfloor k_e \rfloor$ or $\lceil k_e \rceil$ would give a minimal value of $\delta_3$.

**Group 4: Linear parameters size and logarithmic proof size.** Schemes which correspond to the Group 4 are: Dory (Lee, 2021), Gemini (Bootle et al., 2022), Bulletproofs (Bunz et al., 2018) and Compressed Sigma-Protocol (Attema & Cramer, 2020). Taking into account their complexities, we define $\delta_4$ in the following way:

$$\delta_4(k; c, n) = ack + \frac{nb \cdot \log(ck)}{k}$$

Let $r = \frac{a}{bnc}$ and $K = kc$, then

$$\delta_4(k; c, n) = bnc(rK + \frac{\log(K)}{K})$$

One can notice that the optimal value of $K$ depends on the value of $r$: the smaller the value of $r$, the larger the value of the optimal $K$. We present a plot demonstrating the dependence of the optimal $K$ on $r$ in Figure 3 (a). As a result, using this plot, one can choose the ZKP scheme with linear parameters size and logarithmic proof size and infer the optimal value of $k$ with respect to the parameters of the chosen scheme and parameters of the FL setting.

**Group 5: $|C| \log |C|$ parameters size and constant proof size.** Lastly, we describe $\delta_5$ which corresponds to the Group 5 which consist of only one approach: vnTinyRAM (Ben-Sasson et al., 2014).

$$\delta_5(k; c, n) = ack \cdot \log(ck) + \frac{nb}{k}$$

Applying the same substitution as for the $\delta_4$:

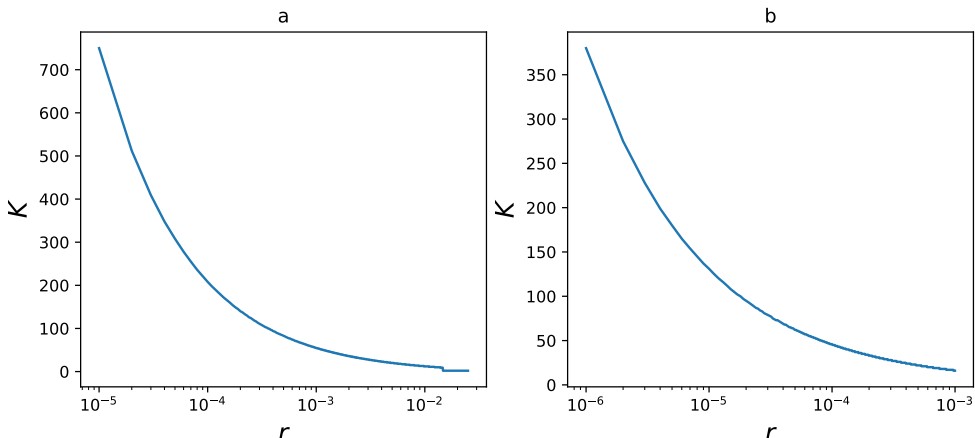

Figure 3: The dependence of the optimal $K$ on $r$ for $\delta_4$ (a) and $\delta_5$ (b).

$$\delta_5(k; c, n) = bnc(rK \cdot \log(K) + \frac{1}{K})$$

Similarly, to the previous group, the plot demonstrating the dependence of the optimal $K$ on $r$ is presented in Figure 3 (b).

With the above generally applicable strategy, one can determine the optimal number of grouped circuits to prove for various ZKP schemes in order to minimize the storage cost or, in other words, communication cost of a verifiable ZKP-based FL protocol.

Still, once the number of circuits in a group is fixed, the costs get linear in the number of groups. In practice, while the simple strategy discussed above already allows one to handle reasonably large learning tasks, there is a maximal size to a group, e.g., due to memory storage limits. Hence, the asymptotic cost given a bound on memory storage remains linear, and given the ever growing size of ML tasks one could wonder whether one can do even better.

One idea which may address this issue, but has not been tried in practical FL settings, is to compose proofs recursively, in particular to use a ZKP scheme that allows for constructing proofs that attest to the correctness of both circuits and other proofs in the same ZKP scheme. Following this idea, at the beginning a prover creates a proof for the first part of computations and then iteratively generates a proof for the correctness of both the next circuit and all preceding proofs. If successful, this idea then could keep the work of the prover about linear in the number of computations, the memory storage of the prover constant, the proof size and communication cost constant and the verifier cost logarithmic.

Several papers have studied the theory of this type of approach. For example, in the context of incrementally verifiable computations (IVC) (Kothapalli et al., 2022; Kothapalli & Setty, 2022; 2023; Bünz & Chen, 2023), proof-carrying data (PCD) (Zhou et al., 2023) and zk Virtual Machine (zkVM) (Liu et al., 2024). State-of-the-art solutions mostly rely on a folding scheme, a primitive that reduces the task of checking two instances of some relation to the task of checking a single instance (Kothapalli et al., 2022). For instance, authors of Nova (Kothapalli et al., 2022) were the first to apply the folding scheme to IVC, resulting in a scheme where the prover's work is linear to the size of the smallest circuit, and the verifier's time and the proof size are logarithmic. Later, other authors presented advanced versions of the protocol such as SuperNova (Kothapalli & Setty, 2022), HyperNova (Kothapalli & Setty, 2023), and ProtoStar (Bünz & Chen, 2023), extending the original approach to more general settings and more efficient arithmetizations.

However, to the best of our knowledge, there are neither works studying an efficient recursive proof composition to verify computations in FL, nor works exploring whether solutions developed for IVC, PCD and zk VM could be adapted to FL. Among all protocols analyzed in Section 3, only in zkFL (Wang et al., 2024) authors apply a recursive ZKP scheme: Halo 2. Nevertheless, they do not discuss the construction of a minimal circuit or recursion properties of the chosen ZKP. Moreover, Halo 2 is an early recursive ZKP scheme with linear verifier cost which is not practical for large ML tasks. Therefore, the impact of recursion possibilities of the scheme on the FL complexities remains unclear.

We believe that applying more efficient recursive proof composition to verify FL computation remains an open research question and can potentially reduce the gap that prevents the use of ZKPs in practical applications.

## 5  Challenges and future directions

While verifiable cross-silo FL protocols is a well studied area with a plenty of solutions, our analysis reveals several challenges which have not been yet addressed by the research community. Below we discuss potential research gaps and future directions. The overview of the key challenges associated with FL participants is illustrated in Figure 4.

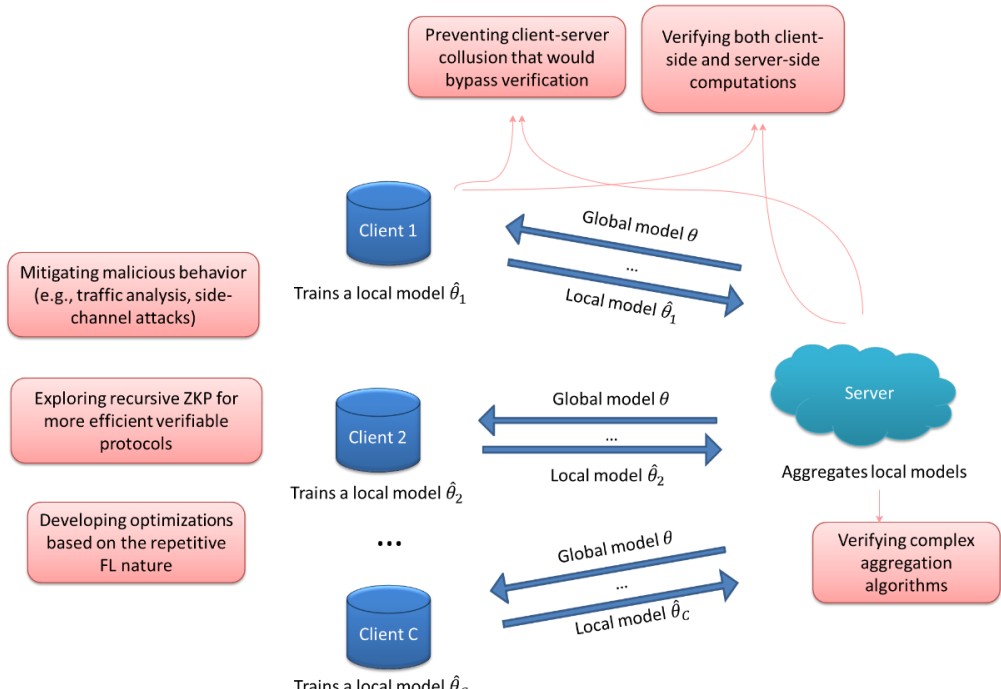

Figure 4: An overview of key challenges and future directions.

Firstly, to the best of our knowledge, there are currently no verifiable FL protocols that fully support verification of both computations performed by clients and server at the same time. Exceptionally, PVD-FL (Zhao et al., 2022) aims to achieve this, however the approach does not guarantee the correctness of all actions of participants and is still vulnerable to poisoning attacks. While some papers, such as (Li et al., 2021), consider threats from both the server and clients, authors do not verify the clients' behavior, but analyze the distribution of their values to address potential attacks. Instead, we believe it would be an interesting direction to design a protocol that would enable verification (following the definition in 2.4) of actions of all participants. It would be later intriguing to compare different approaches from the threat models and the efficiency perspectives.

Secondly, we observe that verifiable aggregation is primarily studied for the most popular type of aggregation – averaging of vectors possessed by DOs. Nevertheless, in certain settings, other U-statistics with a kernel

of the degree two or larger (e.g. Kendall rank correlation coefficient) could be applied (Bell et al., 2020), introducing new challenges in the verification process. Based on our analysis, we can see that ZKP-based verification would potentially fit, however finding an efficient solution still remains an open problem.

Thirdly, we observe that the repetitive nature of FL training is usually overlooked while developing a verifiable protocol. Nevertheless, this property opens up an opportunity to design various optimizations. For instance, in (Guo et al., 2021), the authors considered this property to combine verification of multiple epochs together, thereby reducing the computational cost of clients. Following similar ideas, in Subsection 4.2, we proposed an optimization based on the observation that instead of repetitive proofs in FL, one can group circuits together before proving, thereby optimizing communication costs. However, we believe that other optimizations, for example, for approaches from other taxonomy groups, require further analysis.

Fourthly, among the papers considered in this survey, there are no protocols that are robust against a collusion between client and server to bypass the verification. However, in real world scenarios such collusion might occur. Interestingly, in VerSA (Hahn et al., 2023), the authors do not consider collusion attacks, justifying this choice with the impossibility result shown in (Gordon et al., 2015). The authors of the latter paper demonstrated that in specific settings multi-client verifiable computation cannot be achieved in the presence of users colluding with the server. Nonetheless, the specific setting considered does not necessarily apply to FL. For instance, in contrast to the setting from (Gordon et al., 2015), in a verifiable FL protocol clients may exchange messages with each other or prove computations interactively. As a result, this impossibility result does not necessarily show that a verifiable FL protocol cannot be developed to be robust against server-client collusion attacks. In theory, verification can offer robustness against client-server collusion using zero-knowledge proofs by (a) letting both the clients and the server prove their computations, (b) letting clients and servers show that the hidden intermediate results their proofs refer to (e.g., the messages sent) are consistent, e.g., by proving/publishing identical hash values of these secrets. We believe that the development of a practical protocol is an interesting direction for future research.

We also observe that all verification protocols from our analysis rely either on purely honest, honest-but-curious, or forger threat models, or extend them with robustness against drop-outs. There exist other attacks, to which defending is often harder, e.g., attacks based on side channels or traffic analysis. None of the reviewed approaches formally prove to be robust against any malicious party. Still, the attacks considered in our analysis are likely the most important ones that can afflict the most harm.

Another interesting direction for future research would be to explore existing open-source implementations, available frameworks or tools for verifiable FL. We believe it would be valuable for practitioners to have access to a comprehensive analysis of real-world systems where verification has been successfully implemented.

Lastly, as described in Subsection 4.2, to the best of our knowledge, there are no works exploring the applicability of recursive ZKP schemes in the context of FL. In recent years, there has been an active research in the ZKP community to develop such schemes (Bowe et al., 2019; Kothapalli et al., 2022; Kothapalli & Setty, 2023). We anticipate that they deserve a particular attention. Their characteristics may allow for new optimizations and significant reduction in complexities of verifiable protocols.

# 6 Conclusion

In this paper, we presented a survey on verifiable cross-silo FL. We proposed a new taxonomy distinguishing four categories of verification techniques. We described general design patterns for each category and provided an analysis of threat models, computational and communication costs both per client and per server for each protocol. We also discussed how various features of the cross-silo setting impact the verification process and highlighted advantages of ZKP-based protocols. As a continuation of our conclusions, we discussed the applicability of different ZKP schemes for cross-silo FL and optimization strategies to minimize the communication cost. Finally, we described several research challenges revealed in our analysis and indicated future scientific directions.

**Acknowledgment**

We would like to thank Anca Nitulescu for fruitful discussions on ZKPs. We also thank the action editor and anonymous reviewers for their valuable comments.

This work was partially supported by the 'Chair TIP' project funded by ANR, I-SITE, INRIA and MEL, the Horizon Europe TRUMPET project grant no. 101070038 and has benefited from French State aid managed by ANR under France 2030 program with the reference ANR-23-PEIA-005 (REDEEM project).

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
