# OpenReview forum: "A Survey on Verifiable Cross-Silo Federated Learning"
_TMLR — Accepted by TMLR_

### Review · Reviewer_HNsx · 2025-03-24

**Summary Of Contributions:**

This paper presents a survey on verifiable cross-silo federated learning. The authors introduce some cryptographic backgrounds needed (e.g. blockchain, ZKP) and then provides a taxonomy to classify existing methods according to the threat models (honest, semi-honest, forger, malicious) and the methodology adopted (e.g. ZKP, HP, RA). The authors present an extensive analysis on the complexity (computation, communication) of existing methods.

**Audience:**

Yes

**Broader Impact Concerns:**

No.

**Claims And Evidence:**

Yes

**Requested Changes:**

See 'weaknesses'.

**Strengths And Weaknesses:**

# Strengths:
1. This survey tackles an important topic of verifiable FL, and positions it adequately in the cross-silo setting.
2. This survey presents an extensive and detailed analysis on the threat models and complexities of existing works.

# Weaknesses:
I am not an expert in cryptography so I am not able to evaluate the correctness of this paper. Some minor comments include:
- The authors can outline the overall process of FL + verification with some pseudocode to give readers an idea about when and where the verification process will kick in.
- In the abstract, the authors mention 'FL is a .... with data distributed across multiple devices'. Since this paper tackles cross-silo FL, the authors may revise this statement.

---

> ### Author Response · Authors · 2025-05-13
> **Response to Reviewer HNsx**
>
> Thank you for the feedback.
>
> > The authors can outline the overall process of FL + verification with some pseudocode to give readers an idea about when and where the verification process will kick in.
>
> Verification is not a separate part in the FL process, but often entails some additional work for every calculation allowing to demonstrate at the end some form of correctness. For example, one can use a 9-checksum, which involves some additional calculation in every step (summing the digits to obtain values modulo 9) to be able to verify (with some probability) the result.  E.g., when we calculate $1234 \times 5678+90 = 7006742$, one could do some additional work in every step of the calculation, i.e., $1234 \bmod 9 = 1$, $5678 \bmod 9 = 8$, $90 \bmod 9 = 0$, $7006742 \bmod 9 = 8$ which would allow afterwards to check $(1 \times 8+0) \bmod 9 = 8 \bmod 9$.
> Choosing module 9 is for the ease of calculation of humans, but some verification algorithms we discuss use the same idea, e.g., are based on homomorphic signatures computed on the input so that a check similar to the "modulo 9 check" is possible, e.g., see pseudocode in Algorithm 2.
>
> Additionally, following your suggestion, in our revision we added examples when and where the verification process typically appears (Section 2.4).
>
> >In the abstract, the authors mention 'FL is a .... with data distributed across multiple devices'. Since this paper tackles cross-silo FL, the authors may revise this statement.
>
> In response to your comment, we modified the abstract to maintain the focus on the cross-silo setting.

---

### Review · Reviewer_SDe5 · 2025-04-15

**Summary Of Contributions:**

This paper presents a detailed survey of verifiable cross-silo federated learning (FL) protocols. The authors introduce a taxonomy that groups existing methods into four main categories: redundant aggregation (RA), homomorphic property-based (HP), zero-knowledge proof-based (ZKP), and data embedding (DE). They provide comparative analysis of protocol efficiency and threat models, along with an in-depth section on ZKP schemes and how they can be optimized for FL settings. The paper ends with a discussion of open research questions.

**Audience:**

Yes

**Broader Impact Concerns:**

No major concerns. A short Broader Impact section would help, especially to touch on risks of misuse (e.g., selectively verifying only favorable outcomes), and clarify that verifiability alone doesn’t eliminate all risks (e.g., privacy-compliant attacks).

**Claims And Evidence:**

Yes

**Requested Changes:**

* Important: Add a short discussion on how privacy (e.g. differential privacy) limits what verifiability can detect. This is mentioned briefly but could be supported by recent formal results (see below).

* Important: Include some qualitative discussion of real-world deployment barriers, especially for blockchain-based protocols.

* Moderate: Clarify what types of collusion are out of scope and whether client interaction helps.

* Minor: Provide practical suggestions or rules of thumb for selecting ZKP schemes depending on model complexity and training dynamics.

On the first point above: it may be worth discussing the following:

[AGGPS, ICML '23] "On the Privacy-Robustness-Utility Trilemma in Distributed Learning"

[AKFJG, ICML '24] "The Privacy Power of Correlated Noise in Decentralized Learning"

These works don’t focus on verifiability directly, but are relevant because they formalize how privacy constraints (like differential privacy) limit robustness and detectability of certain attacks (privacy and/or model poisoning). [AGGPS, ICML '23] gives lower bounds that are directly relevant to the point raised in Section 3.1 about privacy limiting the ability to detect poisoning. [AKFJG, ICML '24] studies correlated noise, which can be relevant in scenarios where verifiability relies on assumptions about randomness or independence across clients.

A brief discussion of these results would help ground the survey’s claims about the tradeoffs between verifiability and privacy, and clarify the limits of what’s achievable. This isn’t critical for acceptance but would improve the paper.

**Strengths And Weaknesses:**

**Strengths**

* Clear and useful taxonomy that helps organize a diverse literature.

* Strong technical depth in the comparison of protocol overheads (Tables 1 and 2).

* The ZKP analysis is comprehensive, and the storage optimization section is original and well thought out.

* Identifies real research gaps and future directions.

**Weaknesses**

* The survey doesn't touch much on practical deployment or implementation challenges.

* Limited discussion of blockchain overheads (e.g., miner redundancy, latency, energy).

* Threat model discussion could go deeper, especially around client-server collusion.

* The interaction between verifiability and other goals like privacy and robustness is underexplored.

---

> ### Author Response · Authors · 2025-05-13
> **Response to Reviewer SDe5**
>
> Thank you for your helpful comments.
>
> Following your suggestions we updated the paper as follows :
>
> > Important: Add a short discussion on how privacy (e.g. differential privacy) limits what verifiability can detect. This is mentioned briefly but could be supported by recent formal results (see below).
>
> We extended the discussion in Section 3.1 among other points mentioning robust FL approaches and pointed the reader to the primer on Byzantine ML (the primer discusses the formal results shared in the review).
>
> > Important: Include some qualitative discussion of real-world deployment barriers, especially for blockchain-based protocols.
>
> We updated the discussion in Section 3.3 related to the blockchain-based approaches mentioning the real-world deployment challenges.
>
> >Moderate: Clarify what types of collusion are out of scope and whether client interaction helps.
> >...
> >Threat model discussion could go deeper, especially around client-server collusion.
>
> We have updated the discussion on collusion in Section 5.
>
> >Minor: Provide practical suggestions or rules of thumb for selecting ZKP schemes depending on model complexity and training dynamics.
>
> We have updated Section 4.1 with a discussion on how to select a verification technique for a particular application.

---

> > ### Comment · Reviewer_SDe5 · 2025-06-04
> >
> > Thanks for the update. I think it would be nice to discuss/add the papers I mentioned in my original review instead of the survey you added, since the latter is not focused on privacy.

---

### Review · Reviewer_dGgQ · 2025-05-05

**Summary Of Contributions:**

The paper presents a comprehensive survey of techniques that enable verifiability in cross-silo Federated Learning (FL) systems, where depending on the scenario, participants or the aggregator may not be trusted. Thus, verifiable protocols allow to check that the computations of different entities do not deviate from the training procedure and perform the computations correctly. The contributions of the paper also include a taxonomy of existing methods for FL verifiability, considering the specific challenges of cross-silo settings; a comparison of different Zero Knowledge Proof (ZKP) analyzing their suitability in cross-silo FL; and a discussion of research gaps and challenges.

**Audience:**

Yes

**Claims And Evidence:**

Yes

**Requested Changes:**

+ More detailed justification of the novelty of the survey compared to previous works. For example, what are the differences with previous surveys in FL verifiability with respect of the proposed taxonomy? In which ways the previous surveys are not covering the specific challenges on verifiability in cross-silo FL?
+ More detailed analysis of the threats that are available and those that are mitigated when using verifiability techniques in the server, the participants, or both. For example, in Section 2.3, how verification can prevent data poisoning attacks? If the data has been manipulated before commitment verifiability is of no use for preventing them. On the other hand, if verifiability is only applied in the aggregator, model poisoning attacks could still be possible.
+ Deeper analysis on the privacy vs verifiability trade-off (see previous comments).
+ It would be interesting to include available frameworks or tools for verifiable FL or present some case studies or real-world FL systems using verifiability.

Minor questions:
+ In Section 2.8 the authors say: “In this work, we primarily discuss Non-Interactive Zero-Knowledge (NIZK) proofs”. Why is that? Is this the only approach used for FL verifiability?
+ In Section 4.1, for the CRS it would be convenient to be more explicit about the benefits of having a scheme with a universal reference string.

**Strengths And Weaknesses:**

Strengths:
+ Trust in FL systems is an important aspect, especially in cross-silo settings, where the sensitivity of the tasks and the data in some applications require to have either a trusted aggregator, trusted participants, or both. The paper provides a comprehensive survey of different techniques addressing the specific challenges in cross-silo settings, and advocates for the advantages of some ZPK schemes compared to other approaches for FL verifiability.
+ The taxonomy is very clear, and Figure 1 provides a very nice snapshot of the different group of approaches, techniques, and their intersection. The discussion also includes trade-offs among security, efficiency, and scalability, giving readers a deeper view rather than a superficial comparison. Tables 1 and 2 provide a nice overview of the characteristics of the techniques introduced in the survey. Similarly, the generic algorithms described in the paper are very helpful to understand better how the different families of techniques work.
+ The argument in favor of ZKP-based techniques for verifiability of cross-silo FL seem reasonable and, the analysis in Section 4 provides a comprehensive analysis of the characteristics and requirements needed for applying ZKP techniques in these settings.
+ The authors provided a nice discussion in Section 5, especially identifying some research gaps that can foster future research in this area.

Weaknesses:
+ The paper does not provide a detailed justification in its novelty and contributions with respect to previous surveys in FL verifiability, e.g., (Zhang & Yu, 2022; Tariq et al., 2023). For example, the authors could provide more details on what the novelty of the proposed taxonomy is compared to previous works or be more specific about how previous papers are missing the specific challenges in cross-silo applications.
+ I think that the paper should provide a clearer distinction between different use cases: verifiability on the aggregator, the participants, or both. For instance, this could be introduced in Section 2.4. As this has implications across different aspects (e.g., threat model, scalability requirements, etc.), I think that making the distinction between different use cases clearer would help to improve the analysis and readability of the paper.
+ Following the previous point, the threat models heavily depend on what entities are trusted or not and, depending on the scenarios some type of attacks can be available or not. In this sense, I think the paper lacks a clearer view on what type of attacks are possible depending on where verifiability is applied.
+ While the survey provides a nice conceptual framing, it lacks an empirical analysis, for example comparing some existing methods, which would help the readers to understand better the potential and limitations of these techniques.
+ The paper does not mention or discuss any available frameworks, tools or benchmarks that could help researchers to evaluate verifiable FL techniques. On the other hand, the paper would benefit from discussing case studies or real-world FL systems where verifiability has been successfully implemented (or not).
+ Although the privacy vs verifiability trade-off is acknowledged, it is not explored deeply in the paper. This is very relevant, as privacy can be a key requirement for some cross-silo applications of FL. It would be convenient to explore how to navigate this trade-off with examples or proposed frameworks.

---

> ### Author Response · Authors · 2025-05-14
> **Response to Reviewer dGgQ : Part 1**
>
> Thank you for the valuable feedback.
>
> >More detailed justification of the novelty of the survey compared to previous works. For example, what are the differences with previous surveys in FL verifiability with respect of the proposed taxonomy? In which ways the previous surveys are not covering the specific challenges on verifiability in cross-silo FL?
>
> Regarding the novelty of our work in comparison to prior works [1,2], below we provide a more detailed explanation supporting the claims from sections 1 and 2.4.
>
> The survey by Zhang and Yu [2] is the first effort to systemize different verifiable FL protocols. Their taxonomy categorizes protocols based on the FL setting (centralized/decentralized) and the type of computations to be verified (model updates/aggregation). While their taxonomy helps to describe various works in a structured manner, it does not reflect how the verification is implemented. Consequently, it does not capture many common properties of different verifiable protocols. In contrast, we propose a taxonomy based on the underlying verification techniques. This choice allows us to group similar approaches together and identify common design patterns that directly impact protocol properties (e.g., complexities, threat model assumptions). As a result, we present an analysis that reveals new connections between existing protocols and compare their strengths/limitations.
>
> The survey by Tariq et al. [1] focuses on Trustworthy FL, covering a broad range of topics such as fairness, interpretability, security and privacy. While the verifiability property is also discussed as a part of trustworthiness, the analysis presented is relatively high-level compared to our work. In contrast to [1], we formally describe the underlying verification techniques and present an analysis of the complexities and threat models for considered approaches, which allows us to thoroughly compare protocols providing an in-depth analysis.
>
> Additionally, in contrast to other surveys, we analyze the applicability of different verification techniques in the cross-silo setting, discussing, among other points, which complexity dependencies are critical for practical deployment and which techniques are better suited to realistic threat assumptions. Finally, in Section 4, we discuss the applicability of ZKP for verification in the cross-silo FL that has not been addressed in prior works.
>
> Following your suggestion, in the revision, we have highlighted the key differences in the introduction section in the revised version.
>
>    > More detailed analysis of the threats that are available and those that are mitigated when using verifiability techniques in the server, the participants, or both. For example, in Section 2.3, how verification can prevent data poisoning attacks? If the data has been manipulated before commitment verifiability is of no use for preventing them. On the other hand, if verifiability is only applied in the aggregator, model poisoning attacks could still be possible.
>
> In response to your suggestion regarding the distinction between different use cases, we have added a clarification in Section 2.4. Since participants' roles may vary depending on the specific protocol, our taxonomy focuses on the type of computations to be verified rather than on parties whose actions should be verified. Regarding the question about malicious DOs who poison data before committing to it, we refer to the updated discussion in Section 3.1. To improve clarity, we have added this attack in the text as an example.
>
> > Deeper analysis on the privacy vs verifiability trade-off (see previous comments).
>
> In our revision, we have updated the discussion in Section 3.1 to include a more detailed analysis and to clarify that in many cases verification and privacy can be combined to a large extent.
>
> >It would be interesting to include available frameworks or tools for verifiable FL or present some case studies or real-world FL systems using verifiability.
>
> While this paper surveys approaches studied in literature, it is worth noting that popular open-source libraries such as Flower, NVIDIA FLARE and PySyft typically do not yet include facilities to verify the federated learning they offer to users.
> We have updated Section 5 to include this suggestion as an interesting direction for future research.
>
>
> [1] Zhang & Yu, 2022;
> [2] Tariq et al., 2023;

---

> ### Author Response · Authors · 2025-05-14
> **Response to Reviewer dGgQ : Part 2**
>
> Minor questions:
>
>  > In Section 2.8 the authors say: “In this work, we primarily discuss Non-Interactive Zero-Knowledge (NIZK) proofs”. Why is that? Is this the only approach used for FL verifiability?
>
> In the considered papers only non-interactive ZKP were applied, thus we relied on the corresponding definition for convenience. In theory, interactive scheems could be applied, but they are often less efficient than non-interactive ones and therefore less popular.
>
>  >In Section 4.1, for the CRS it would be convenient to be more explicit about the benefits of having a scheme with a universal reference string.
>
> Following your suggestion, in the revision we have updated Section 4.1 to mention the benefits of a universal CRS.

---

### Review · Reviewer_az79 · 2025-05-08

**Summary Of Contributions:**

This paper presents a comprehensive servey on verifiable FL in the context of context of cross-silo FL. The authors introduce a taxonomy that classifies existing approaches into four categories: Redundant Aggregation (RA), Homomorphic Property (HP), Zero-Knowledge Proof (ZKP), and Data Embedding (DE). The comparative analysis of ZKP with the focus on cost and scalability in FL contexts is also provided.

**Audience:**

Yes

**Broader Impact Concerns:**

No.

**Claims And Evidence:**

Yes

**Requested Changes:**

- Add more visual figures or diagrams to summarize key comparisons and findings.
- Improve the connection between the ZKP optimization analysis and the other part of the survey.

**Strengths And Weaknesses:**

Strengths:

- The survey is well written and easy to understand
- The analysis of ZKP for verifiable FL is very detailed
- The proposed classification into four categories is clear and sound.

Weaknesses:

- The theoretical modeling and optimization of ZKPs feel somewhat disconnected from the rest of the paper. Strengthening the connection between these sections and the earlier taxonomy would improve the narrative flow.
- The paper includes a large number of detailed tables, but only a single main figure. More visual summaries (e.g., comparative diagrams) would enhance readability.
- There is some overlap between the taxonomy section and the subsequent ZKP analysis, which could be streamlined to reduce redundancy.
- The scope may be too narrow. While the focus on cross-silo settings is well motivated, including a brief comparison with cross-device FL could improve the generalizability of the survey.

---

> ### Author Response · Authors · 2025-05-19
> **Response to Reviewer az79**
>
> Thanks for your comments and suggestions!
>
> > Add more visual figures or diagrams to summarize key comparisons and findings.
>
> In our revision, we have added new figures to support key elements of our discussion with relevant illustrations. In Figure 2, we present a timeline of considered approaches with respect to the taxonomy categories. In Figure 4, we illustrate the main challenges and future directions summarized in Section 5.
>
> > Improve the connection between the ZKP optimization analysis and the other part of the survey.
>
> While describing various verifiable FL approaches, we observed that ZKP-based methods offer several powerful features (summarized in the last paragraph of Section 3) that are not available via other verification techniques. Consequently, our analysis naturally led to a dedicated subsection focused on the ZKP-based methods. To ensure a smooth transition between the overview of approaches and the optimization analysis, we discussed the applicability of ZKPs and the impact of their characteristics within the context of FL in Section 4.1. In our revision, we also added a discussion on how to select a ZKP scheme for a given application, which we believe further smooths the transition between different parts of our survey.

---

### Decision · Action_Editor_2Rc3 · 2025-06-10

**Recommendation:** Accept as is

**Additional Comments:**

Small remark for the camera ready: please go through the mathematical formulas and consider writing $\log$ instead of $log$, $\max$ instead of $max$ etc.

**Audience:**

Yes

**Audience Explanation:**

As also agreed by the reviewers, this survey is probably interesting to many in the TMLR audience.

**Claims And Evidence:**

Yes

**Claims Explanation:**

As summarized in the reviews, the paper classifies existing verifiable FL approaches into four categories based on the nature of the methods: redundant aggregation, homomorphic property, zero-knowledge proof, and data embedding. For ZKP there are also comparisons of the existing methods in terms of cost and scalability. The paper seems like a valuable introduction to this topic for general audience of TMLR.

All the reviewers agree about the value of this review work on verifiable cross-silo FL, and all are leaning towards acceptance.